# Health Services Use and Health Outcomes among Informal Economy Workers Compared with Formal Economy Workers: A Systematic Review and Meta-Analysis

**DOI:** 10.3390/ijerph18063189

**Published:** 2021-03-19

**Authors:** Nisha Naicker, Frank Pega, David Rees, Spo Kgalamono, Tanusha Singh

**Affiliations:** 1National Institute for Occupational Health, A Division of the National Health Laboratory Service, Johannesburg 2001, South Africa; SpoK@nioh.ac.za (S.K.); TanushaS@nioh.ac.za (T.S.); 2School of Public Health, University of the Witwatersrand, Johannesburg 2193, South Africa; DavidR@nioh.ac.za; 3Department of Environmental Health, University of Johannesburg, Johannesburg 2028, South Africa; 4Department of Environment, Climate Change and Health, World Health Organization, 1211 Geneva, Switzerland; pegaf@who.int; 5Department of Clinical Microbiology and Infectious Diseases, School of Pathology, University of the Witwatersrand, Johannesburg 2193, South Africa

**Keywords:** occupational health, health inequalities, informal economy, health services use, occupational injuries, depression

## Abstract

*Background:* There are approximately two billion workers in the informal economy globally. Compared to workers in the formal economy, these workers are often marginalised with minimal or no benefits from occupational health and safety regulations, labour laws, social protection and/or health care. Thus, informal economy workers may have higher occupational health risks compared to their formal counterparts. Our objective was to systematically review and meta-analyse evidence on relative differences (or inequalities) in health services use and health outcomes among informal economy workers, compared with formal economy workers. *Methods:* We searched PubMed and EMBASE in March 2020 for studies published in 1999–2020. The eligible population was informal economy workers. The comparator was formal economy workers. The eligible outcomes were general and occupational health services use, fatal and non-fatal occupational injuries, HIV, tuberculosis, musculoskeletal disorders, depression, noise-induced hearing loss and respiratory infections. Two authors independently screened records, extracted data, assessed risk of bias with RoB-SPEO, and assessed quality of evidence with GRADE. Inverse variance meta-analyses were conducted with random effects. *Results:* Twelve studies with 1,637,297 participants from seven countries in four WHO regions (Africa, Americas, Eastern Mediterranean and Western Pacific) were included. Compared with formal economy workers, informal economy workers were found to be less likely to use any health services (odds ratio 0.89, 95% confidence interval 0.85–0.94, four studies, 195,667 participants, I^2^ 89%, low quality of evidence) and more likely to have depression (odds ratio 5.02, 95% confidence interval 2.72–9.27, three studies, 26,260 participants, I^2^ 87%, low quality of evidence). We are very uncertain about the other outcomes (very-low quality of evidence). *Conclusion:* Informal economy workers may be less likely than formal economy workers to use any health services and more likely to have depression. The evidence is uncertain for relative differences in the other eligible outcomes. Further research is warranted to strengthen the current body of evidence and needed to improve population health and reduce health inequalities among workers.

## 1. Introduction

Globally, approximately two billion (61%) workers work in the informal economy [1], including workers in informal sectors and those in the formal economy but in informal work arrangements [2]. This labour force, compared with the formal economy’s, is often marginalised, covered incompletely or not at all by health and safety regulations and labour laws, and has no or limited access to social protection, especially in low to middle income countries [3]. Consequently, their working environments may have poor hazard control, and they may experience greater occupational health risks than their counterparts in the formal economy [3]. The World Health Organization, including through its global Commission on Social Determinants of Health, has identified informal economy workers as a key population for action on the social determinants of health to improve health equity [4].

Health services use and health outcomes may differ between informal and formal economy workers, indicating differences (or inequalities) in health among workers by formality of the economy. The former often face multiple barriers to use, including long working hours, high opportunity costs (e.g., loss of income taking time off work), unaffordability and remote health services. Migrant undocumented workers may not feel welcome at health facilities [5,6]. Additionally, informal economy workers may have less access to occupational health education and services; resulting in less knowledge of hazards and their control, poor access to personal protective equipment, and low health services coverage [7,8,9].

Despite limited research on the informal economy, many adverse health outcomes have been demonstrated among this population, including cancer, traumatic injury, respiratory disease, noise induced hearing loss, musculoskeletal disorders, communicable and mental diseases [10,11,12,13,14].

Although research is available on health services use and health outcomes, the degree to which these differ by formality of work remains unclear. To our knowledge, there are no previous systematic reviews and meta-analyses of health services use and health outcomes among informal economy workers, compared with their formal economy counterparts. Such evidence is needed for policy formulation and regulation; and to design, plan, cost, implement and evaluate interventions that improve health equity.

In this article, we present a systematic review and meta-analysis of health services use and health outcomes among informal economy workers compared with those in the formal economy.

## 2. Materials and Methods

This systematic review used the Preferred Reporting Items for Systematic Reviews and Meta-analyses (PRISMA) guidelines [15]. The protocol was registered on PROSPERO (CRD42018108894).

### 2.1. Literature Search

A search of Ovid Medline, PubMed and Embase was conducted in March 2020. The PubMed search strategy, presented in Appendix A, was adapted for the other databases. Reference lists of articles included in the study were hand-searched for relevant additional studies. Experts from the WHO, International Labour Organization (ILO) and WHO Collaborating Centres for Occupational Health were requested to identify any additional published and unpublished studies potentially eligible for this systematic review.

### 2.2. Eligibility Criteria

The eligibility criteria were informed by relevant PECO criteria [16] and are described below.

#### 2.2.1. Types of Populations and Exposures

We included informal economy workers, defined as informal employers, informal own-account workers, informal wage workers and informal wage workers in households [17]. We included studies of adult workers (≥16 years based on ILO criteria) in these four categories, in any economic sector or occupational group. The relevant exposure was working in the informal economy.

#### 2.2.2. Types of Comparators

As comparators, we included workers in the formal economy, or the total population of a country or subnational geographic unit (e.g., district or city) as a proxy for formal economy workers.

#### 2.2.3. Types of Outcomes

An expert group of occupational medicine, public health medicine, epidemiology and health equity specialists with experience in exposures and health outcomes in occupational settings selected the ten health outcomes they judged most relevant for health of workers in the informal economy (Table 1). Almost all of these outcomes are aligned with sustainable development goals indicators (Table 1).

#### 2.2.4. Types of Studies

Studies of any quantitative design, comparing informal with formal economy workers in any county, were eligible. We excluded qualitative studies, case reports, modelling studies, study records without quantitative data (e.g., commentaries and perspectives), and studies that compared groups of informal economy workers (e.g., inter-country investigations).

Studies carried out and published between 1 January 1999 and 30 March 2020 in any language were included, provided they had an English abstract.

We included studies with estimates of relative differences (e.g., a hazard, risk or odds ratio) between informal and formal economy workers on an included outcome. These estimates are commonly called measures of relative inequality [18]. For studies providing measures of absolute differences only, we converted these into relative differences (if feasible). If a study presented data for multiple years (e.g., 2009 and 2010), we prioritized the latest data for inclusion in the review (i.e., in this example: 2010).

### 2.3. Data Extraction

The following information was extracted from included studies: first author, year of publication, year of data collection, study setting, study design, industrial sector, occupation, age of participants, formality of economy, outcome assessed, number of cases and non-cases in the exposed (informal economy) and unexposed (formal economy) groups, adjustment for confounding, and the point estimate with confidence intervals (Table 2).

Two review authors independently extracted data, with a third review author resolving conflicting extractions.

### 2.4. Assessment of Risk of Bias

We used a modified version of the RoB-SPEO tool [19], as applied in Hulshof et al. [20], to assess risk of bias in each study by the following domains: selection bias, lack of blinding, exposure misclassification, outcome misclassification, incomplete exposure data, incomplete outcome data, selective reporting of exposures, selective reporting of outcome, differences in numerator and denominator, conflict of interest, and other bias. For each domain, we applied the standard RoB-SPEO ratings of: “low risk”, “probably low risk” “probably high risk”, “high risk” or “no information”. Two review authors independently assessed risk of bias for each study, and a third resolved conflicting ratings. Risk of bias assessments (rating plus justification for selected ratings) are reported in Appendix A (Risk of Bias Table per Study), and a summary is provided in Table 3 (Risk of Bias).

### 2.5. Evidence Synthesis

Two authors independently assessed whether included studies reporting the same outcome were sufficiently homogenous in population, comparator and outcome to potentially be combined in a meta-analysis, with a third author resolving differing opinions. We used Review Manager (RevMan) Version 5.3. for meta-analyses.

We used the inverse variance method and applied random-effects models if we believed that the included studies estimated different, yet related, effects. The statistical heterogeneity of studies was assessed with the I^2^ statistic. We anticipated serious statistical heterogeneity, and thus reported pooled effect estimates from meta-analyses even when high statistical heterogeneity was observed. However, we did not report any pooled estimates when I^2^ > 90%. We were unable to produce funnel plots as planned due to the small number of studies. We narratively synthesised studies that were not suitable for meta-analysis.

### 2.6. Quality of Evidence Assessment

For each outcome, we assessed the quality (or certainty) of its body of evidence, using GRADE (Grading of Recommendations Assessment, Development and Evaluation) [21]. Our assessment considered risk of bias, inconsistency, indirectness, imprecision, and size of OR estimates (i.e., measures of relative inequality [19]). We applied the standard GRADE ratings “high”, “moderate”, “low” and “very low”. Starting at “high”, we downgraded by one level for serious concerns and by two levels for very serious concerns for each domain. Evidence could be upgraded by one level and two levels if the estimated ORs (and thus relative inequality in the outcome) were high (≥2.5) and very high (≥5.0), respectively. However, once downgraded, upgrading was no longer considered. The “Summary of findings” table shows quality of evidence ratings and their justification.

## 3. Results

### 3.1. Study Selection

Twelve studies fulfilled the inclusion criteria (Figure 1).

### 3.2. Characteristics of Included Studies

The characteristics of included studies are presented in Table 2. Of the 12 included studies, nine (75%) originated from the Americas, [22,23,25,26,27,28,30,32,33] two (17%) from Africa [29,31] and one (8%) from the Western Pacific region. [24] The sample size ranged from 114 to 1,411,048 (Table 2). Most studies (83%) had both female and male participants.

Of the included studies, four analysed health services use, [22,23,24,25] but none studied occupational health services use. Two assessed fatal injuries [26,27] and three non-fatal injuries. [28,29,30] Two focused on depression and anxiety and another on depression only [22,31,32]. One paper assessed musculoskeletal disorders, [33] while no papers reported on NIHL, upper and lower respiratory infections, tuberculosis or HIV.

### 3.3. Risk of Bias

Our complete assessments of risk of bias by domain for each study are presented in Appendix A, and the summary of risk of bias in Table 3. We rated all studies as at “probably low” risk of bias due to lack of blinding and conflict of interest. Only two studies received a rating of “definitely high” risk of bias for a domain [26,27]: Lopez-Bonilla et al. (2011) for selective reporting of outcome; and Mora (2011) for incomplete reporting of outcome. For the selection bias domain, five studies were judged as having “probably high” risk of bias [24,28,29,32,33]. Three studies were rated as “probably high” for exposure misclassification [24,32,33] and two studies as “probably high” for outcome misclassification [26,29]. Two studies had “probably high” risk of bias for the differences in the numerator and denominator domain [28,29].

**Table 3 ijerph-18-03189-t003:** Summary of the risk of bias.

Included Study	Selection Bias	Lack of Blinding	Exposure Misclassification	Outcome Misclassification	Incomplete Exposure Data	Incomplete Outcome Data	Selective Reporting of Exposures	Selective Reporting of Outcome	Differences in Numerator and Denominator	Conflict of Interest
	Any health services use									
Giatti 2008 * [22]	+	+	+	+	+	+	+	+	++	+
Giatti 2011 [23]	+	+	+	+	+	+	+	+	++	+
Le 2015 [24]	-	+	-	+	+	+			++	+
Miquillan 2013 [25]	++	+	+	+			+	+	++	+
	Fatal occupational injuries									
Lopez- Bonilla 2011 [26]	+	+	++	-		-	+	--	+	+
Mora 2011 [27]	+	+	+	+		--	+	-	+	+
	Non-fatal occupational injuries									
Cunningham 2012 [28]	-	+	++	+	++	-	++	-	-	+
Calys-Tagoe 2017 [29]	-	+		-	++	-	++	+	-	+
Santana 2003 [30]	+	+	+	++	+	+	+	-	++	+
	Depression									
Abbas 2013 [31]	+	+	++	++	++		++	+	++	+
Da Silva 2006 a [32]	-	+	-	+	+	++	+	++	+	+
Giatti 2008 * [22]	+	+	+	+	+	+	+	+	++	+
	Musculoskeletal disorders									
Da Silva 2006 b [33]	-	+	-	+	++	+	+	+	+	+

RoB-SPEO [19] risk of bias ratings: definitely low (++ dark green); probably low (+ light green); probably high (- pink); definitely high (-- red); no information (yellow). * This study reported evidence on two outcomes: any health services use and depression.

### 3.4. Findings

#### 3.4.1. Any Health Services Use

Four studies with a total of 195,667 participants from two WHO regions (Americas and Western Pacific) reported estimates of relative differences in the likelihood of having used any health services among informal economy workers, compared with formal economy workers [22,23,25,31]. We considered all four studies to be sufficiently homogenous clinically to be combined in the same quantitative meta-analysis. Compared with formal economy workers, informal economy workers were less likely to have used any health services (odds ratio (OR) 0.89, 95% CI 0.85–0.94, 4 studies, 195,667 participants, I^2^ 89%). (Figure 2).

We downgraded by one grade each for serious concerns for inconsistency (I^2^ = 89%) and indirectness (restricted geographical representation and gender). Overall, the quality of this body of evidence was downgraded by two grades, from “high quality” to “low quality”. Informal workers may be less likely to use any health services, compared with formal economy workers. Further research is very likely to have an important impact on our confidence in the conclusion and is likely to change it (Appendix A).

#### 3.4.2. Fatal Occupational Injuries

Two studies with a total of 2,081,543 participants from one WHO region (Americas) reported estimates of relative differences in the risk of having died from an occupational injury among informal economy workers, compared with formal economy workers. We considered these studies to be sufficiently homogenous clinically to be combined in the same quantitative meta-analysis. Because we judged this body to be of “very-low quality”, we did not present a total in the forest plot and narratively synthesised the body of evidence. One study reported a substantially decreased odds of having any non-fatal injury among informal economy workers, compared with formal economy workers (Figure 3) [26]. The other study reported a similar risk among informal and formal economy workers, but with a very wide 95% CI (Figure 3) [27].

We downgraded for very-serious concerns in the risk of bias domain by two grades and by one for each of inconsistency, serious indirectness and imprecision. The quality of evidence was therefore downgraded from “high quality” by five grades to “very-low quality”. We are very uncertain about this outcome among informal economy workers, when compared with formal economy workers.

#### 3.4.3. Non-Fatal Occupational Injuries

Three studies with a total of 3465 participants from two WHO regions (Americas and Africa) reported estimates of relative differences in the risk of having a non-fatal injury among informal economy workers, compared with formal economy workers [28,29,30]. Because we judged this body to be of “very-low quality”, we do not present a total in the forest plot. (Figure 4). One study reported very highly increased odds for this outcome (OR 4.43, 95% CI 2.77 to 7.09) among informal economy workers, compared with formal economy workers [31]; however, the other two studies reported very uncertain estimates with point estimates below one [28,30].

The quality of evidence was downgraded for all GRADE domains. We had serious concerns for risk of bias (Table 3), inconsistency (I^2^ = 94%), indirectness (restricted geographical locations, restricted occupational groups) and imprecision (95% CI crossed 1). Without further research we are unable to draw conclusions about relative differences in this outcome by formality of work. (Appendix A.)

#### 3.4.4. Depression

Three studies with a total of 26,260 participants from two WHO regions (Americas and Africa) reported estimates of relative differences in the risk of having depression among informal economy workers, compared with formal economy workers [22,31,32]. One study assessed depression only [22] (as per our a priori outcome definition), while two studies also assessed depression, but together with anxiety [31,32] (proxy for our outcome definition). A subgroup analysis by outcome definition indicated no evidence for differences by subgroups defined by the outcome definition (depression only vs. depression and anxiety; *p* = 0.17; Appendix A). We considered all three included studies to be sufficiently homogenous clinically to be combined in the same quantitative meta-analysis. Compared with formal economy workers, informal economy workers were more likely to have had depression (OR 5.02, 95% CI 2.72–9.27, 3 studies, 26,260 participants, I^2^ 87%, Figure 5).

The quality of evidence was downgraded by two levels to “low quality of evidence” due to serious concerns for indirectness and imprecision. Since the domains were downgraded by more than two levels, we did not upgrade even though the estimate was high (OR 5.02) (Appendix A). Compared with formal economy workers, informal economy workers may be more likely to have depression.

#### 3.4.5. Musculoskeletal Disorders

Musculoskeletal abnormalities among ragpickers (waste recyclers) were reported in one study compared with workers living in the same neighbourhood [33]. There were three outcome definitions for musculoskeletal disorders (lower back pain, upper limb pain and lower limb pain). We prioritised lower back pain (LBP) as it carries a larger burden of disease and is often the most common site for musculoskeletal pain.^31^ The prevalence of LBP (OR 1.1; 95% CI 0.9–1.1) was similar in both groups. This study had a “probably low” risk for bias overall. However, it was downgraded by two levels for very serious concerns for indirectness because of restricted geographical location and occupational group, as well as by one level for very serious concerns for imprecision (Appendix A). We are very uncertain about this outcome among workers in the informal economy, compared with those in the formal economy.

## 4. Discussion

### 4.1. Summarised Findings

To our knowledge, this is the first systematic review assessing relative differences (or inequalities) in health services use and health outcomes among informal economy works compared with formal economy workers. Informal economy workers may be less likely to use any health services and more likely to have depression than their formal economy counterparts (Table 4). We are uncertain about differences in the other outcomes reviewed (occupational health services use; fatal and non-fatal occupational injury; HIV; tuberculosis; musculoskeletal disorders; NIHL; and respiratory infections) between the two groups because of very-low quality of evidence or absence of studies.

### 4.2. Comparison with Other Evidence

Being the first systematic review on the topic, our findings were compared with other empirical studies. Our findings suggest lower health services use among workers in the informal than in the formal economy, which is not unexpected considering the loss of income while obtaining health care, and typically absent health insurance or health benefits associated with informality. However, the four included studies did not assess specific occupations or industries. Furthermore, Giatti et al. [22,23] studied men only, which is a limitation as health services use may differ between men and women [34,35]. This is underscored by another study that showed women in the formal economy consulted health services providers more [22], but use by both sexes in the informal economy was low. While Le et al. did not find a significant association between sex and health services use [24], the study had a “probably high” risk of bias. Most studies were conducted in Brazil, with one from Asia, thus limiting the generalizability of the results.

The risk of fatal injuries was lower in the informal than the formal economy, but not significantly so. This was based on just two studies, both with high risk of bias. This finding is possibly due to lower ascertainment of these incidents in the informal relative to the formal economy. Alternatively, it is possible that the formal sector utilizes heavier mechanisation, a potential cause of fatal injuries. The contrast between large-scale formal farming and subsistence farming using manual methods supports this theory [36,37]. Also, risk differences may be site and sector specific: informal mining may be riskier relative to regulated formal mining [38,39,40].

Higher risks of non-fatal injuries were non-significantly associated with informality, but not consistently as only one study found an increased risk. The considerations about site and sector specifics are probably pertinent here as well. Secondary prevention to reduce impairment caused by inadequately treated injures could be a serious factor in the informal economy. It is possible that there is a higher risk of non-fatal injuries associated with informality, and underutilisation of health care for these injuries maybe linked to greater impairment than in more formal workplace settings. However, research in this area is lacking.

The higher risks of depression in informal work settings is unsurprising. Characteristics of informality, such as insecure work and financial insecurity, are associated with increased rates of mental disorders [41,42]. The relative underuse of health services by this group, along with their higher risks of depression, is of concern.

### 4.3. Limitations and Research Considerations

This research identified several limitations of the current bodies of evidence on the included outcomes. Exposure misclassification due to poor distinction between formal and informal populations is likely (Table 3). In several studies the “formal” comparator workers probably included some informal workers (this would be particularly likely where neighbourhood controls were used).

All studies were cross-sectional, and other analytic study designs should be considered. Poor mental health or inability to work in the formal economy are factors that may drive people to informal work, creating spurious associations in cross-sectional study designs. Cohort investigations in informal settings are challenging. However, if the workforce is reasonably stable, relatively common acute outcomes (e.g., traumatic injury) could be investigated prospectively using participatory research methods for adequate response rates [43] and retention.

Secondary prevention to reduce impairment caused by inadequately treated injuries in the informal economy needs further investigation. In particular, it would be useful to research whether higher risks of non-fatal occupational injuries are associated with informality; and if underuse of health services for these injuries is linked to greater impairment than in more formal settings. Research methods that control for under ascertainment of fatal accidents, especially in unregistered and uninsured workers, need to be used. In addition, larger studies examining the causes of death by site and industrial sector in the informal relative to the formal economy are needed.

Only 12 studies were suitable for this review despite the wide range of eligible outcomes, and no more than four studies were available to inform the review on any of the outcomes. Research in diverse settings across greater geographic regions is encouraged.

## 5. Conclusions

This is the first systematic review synthesising evidence on health services use and work-related health outcomes by economic formality. Among workers in the informal economy, the review found lower health services use and higher risk of depression. The results did not demonstrate significant differences in fatal or non-fatal occupational injuries and musculoskeletal disorders between informal and formal economy workers. In addition, no eligible study was found for the outcomes of occupational health services use, HIV, tuberculosis, NIHL and respiratory infections. This study highlights the areas of health services and research that are needed to be improved to promote the population health of workers and reduce health inequalities.

## Figures and Tables

**Figure 1 ijerph-18-03189-f001:**
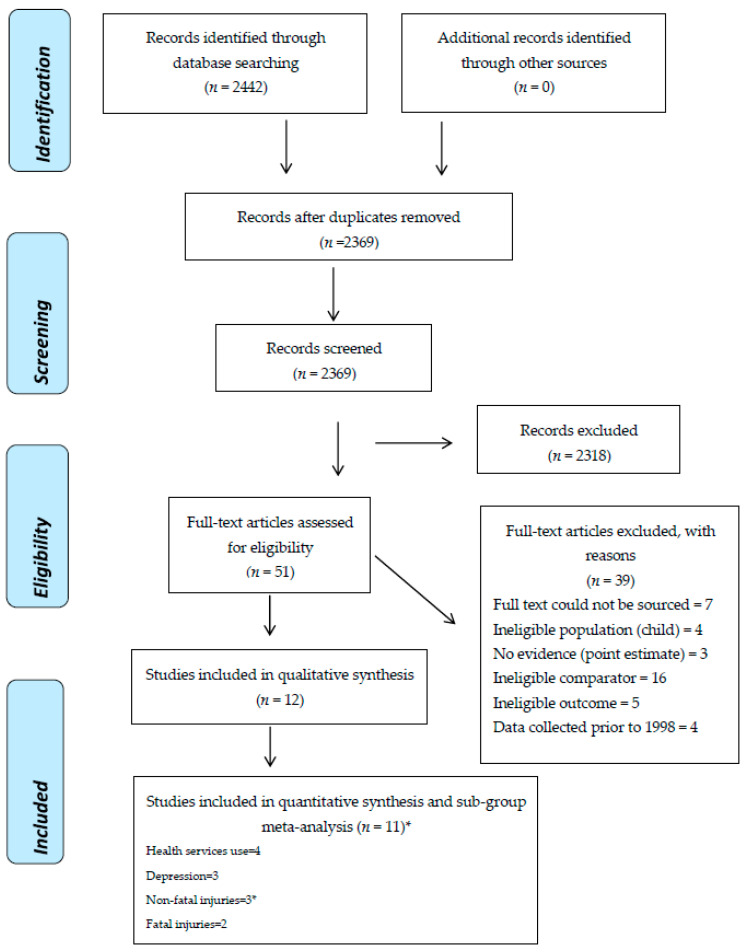
PRISMA diagram of study selection. * One paper included two outcomes.

**Figure 2 ijerph-18-03189-f002:**
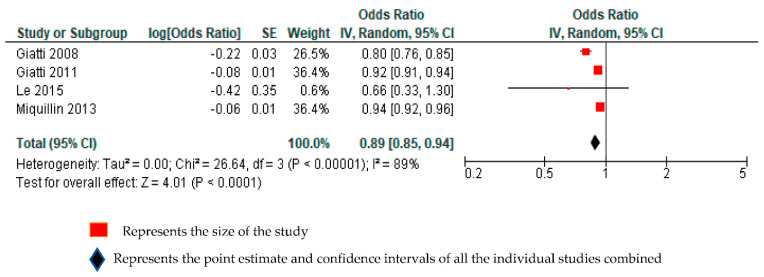
Forest plot assessing any health services use among informal economy workers, compared with formal economy workers.

**Figure 3 ijerph-18-03189-f003:**
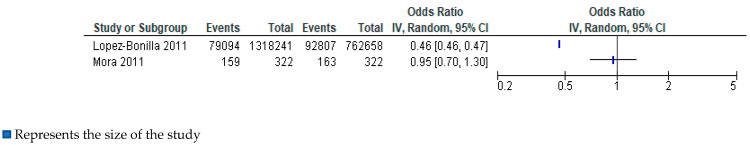
Forest plot assessing fatal occupational injuries in informal economy workers, compared with formal economy workers.

**Figure 4 ijerph-18-03189-f004:**
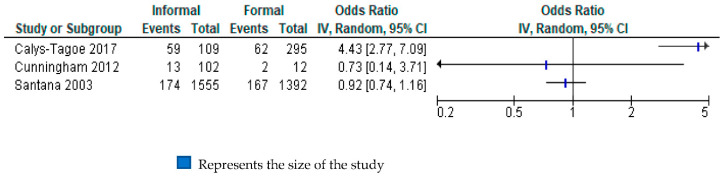
Forest plot assessing non-fatal occupational injuries in informal economy, compared with formal economy workers.

**Figure 5 ijerph-18-03189-f005:**
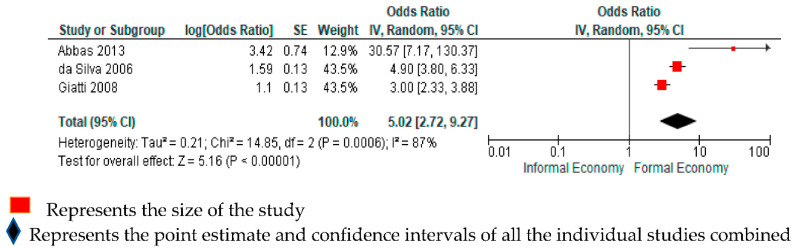
Forest plot for studies comparing depression in informal economy workers, compared with formal economy workers.

**Table 1 ijerph-18-03189-t001:** Prioritized outcomes and related sustainable development goals indicators.

No	Outcome in This Review	Relevant Sustainable Development Goals Indicator
1	Has used any health service	3.8.1 Coverage of essential health services (defined as the average coverage of essential services based on tracer interventions that include reproductive, maternal, newborn and child health, infectious diseases, non-communicable diseases and service capacity and access, among the general and the most disadvantaged population)
2	Has used any occupational safety and health service	
3	Has died from an occupational injury	8.8.1 Frequency rates of fatal and non-fatal occupational injuries, by sex and migrant status
4	Has had any non-fatal occupational injury	
5	Has human immunodeficiency virus infection	3.3.1 Number of new HIV infections per 1000 uninfected population, by sex, age and key populations
6	Has tuberculosis	3.3.2 Tuberculosis incidence per 1000 population
7	Has depression	3.4.2 Suicide mortality rate
8	Has any musculoskeletal disorder	-
9	Has noise induced hearing loss (NIHL)	-
10	Has respiratory infections	3.4.1 Mortality rate attributed to cardiovascular disease, cancer, diabetes or chronic respiratory disease

**Table 2 ijerph-18-03189-t002:** Characteristics of the 12 studies included in the systematic review.

No	First Author	Year of Publication	Country	Study Design	Year of Data Collection	Outcome of Interest	Population	Sex	Occupation	Total Workers in Informal Economy (IE)	Number of Cases in IE	Number of Non-Cases in IE	Total Workers in Formal Economy (FE)	Number of Cases in FE	Number of Non-Cases in FE	Point Estimate (SE) [Ref FE Workers]	95% CI
1	Giatti [22]	2008	Brazil	Cross sectional	2003	Health services use	32,887	Male	Multiple	8255	3599 *	4656 *	16,673	9554 *	7119 *	OR 0.60 (0.03)	0.56–0.64
2	Giatti [23]	2011	Brazil	Cross sectional	2008	Health services use	31,331	Male	Multiple	10,185	5052	5133	21,146	13,089	8057	OR 0.83 (0.05)	0.81–0.85
3	Le [24]	2015	Vietnam	Cross sectional	x	Health services use	1800	Both	Multiple	210	120	90	340	213	127	OR 0.38 (0.35)	0.19–0.74
4	Miquillan [25]	2013	Brazil	Cross sectional	2008	Health services use	152,233	Both	Multiple	62,612	54,347	8265	76,246	65,267	10,979	OR 0.86 (0.01)	0.84–0.89
*5	Lopez- Bonilla [26]	2011	Nicaragua	Cross sectional	2005	Fatal occupational injuries	2,080,899	Both	Multiple	1,318,241	79,094	1,239,147	762,658	92,807	669,851	OR 0.49	
*6	Mora [27]	2011	Costa Rica	Cross sectional	2005/2006	Fatal occupational injuries	x	Both	Multiple	x	159	x	x	163	x	OR 1.05	
*7	Cunningham [28]	2012	Paraguay	Cross sectional	2009	Non-fatal occupational injuries	114	Both	Waste recyclers	102	13	89	12	2	10	OR 0.76	
8	Calys-Tagoe [29]	2017	Ghana	Cross sectional	2014	Non-fatal occupational injuries	404	Both	Small Scale Miners	109	59	50	295	62	233	OR 0.64	0.32–1.18
9	Santana [30]	2003	Brazil	Cross sectional	2000	Non-fatal occupational injuries	2947	Both	Multiple	1555	174	1381	1392	167	1225	OR 0.92	0.74–1.16
10	Abbas [31]	2013	Egypt	Cross sectional	2012	Depression	451	Both	Cleaners	242	143	99	209	62	147	OR 3.4	2.27–5.17
11	Da Silva [32]	2006 a	Brazil	Cross sectional	2004	Depression	881	Both	Waste recyclers	441	197	244	440	148	292	OR 1.4	1.2–1.7
1	Giatti [22]	2008	Brazil	Cross sectional	2003	Depression	32,887	Male	Multiple	8255	x	x	16,673	x	x	OR 1.1	0.87–1.39
12	Da Silva [33]	2006 b	Brazil	Cross sectional	2004	Musculoskeletal disorders	254	Both	Waste recyclers	441	61	380	44	25	415	PR LBP 1.1 PR ULP 1.1 PR LLP 0.9	0.9–1.1 1.0–1.3 0.8–1.1

* Studies 5–7 do not have 95% CI or *p* values.

**Table 4 ijerph-18-03189-t004:** Summary of findings: use of health services and health outcomes among informal economy workers, compared with formal economy workers.

Population: Informal Economy WorkersSetting: Any Country, Occupation, Industrial Sector and WorkplaceComparator: Formal Economy Workers in the Same Country
Outcomes	Anticipated Absolute Risk * (95% CI)	Relative Difference (95% CI)	No of Participants (No of Studies)	Quality of the Evidence(GRADE)	Comments
Risk among Formal Economy Workers	Risk among Informal Economy Workers
Has used any health services	770 per 1000	749 per 1000(740 to 759)	OR 0.89(0.85 to 0.94)	195667(4 studies)	⊕⊕⊝⊝Low—^a,b,c^	Informal economy workers may be less likely to have used any health service, compared with formal economy workers.
Has used any occupational safety and health services	-	-	-	-	-	No evidence available on this outcome.
Has died from an occupational injury	-	-	-	15650750(2 studies)	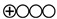 Very low—^a,b,d,e^	We are very uncertain about the estimate for this outcome.
Has had any non-fatal occupational injury	-	-	-	3465(2 studies)	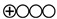 Very Low—^a,b,d,f,g^	We are very uncertain about the estimate for this outcome.
Has depression	20 per 1000	92 per 1000(52 to 158)	OR 5.02(2.72 to 9.27)	26260(3 studies)	⊕⊕⊝⊝Low—^b,g^	Informal economy workers may be more likely to have depression, compared with formal economy workers.
Has any musculoskeletal disorder	-	-	-	881(1 study)	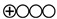 Very Low—^b,g^	We are very uncertain about the estimate for this outcome.

^a.^ Serious concerns for inconsistency due to high statistical heterogeneity (I^2^ > 90%). ^b.^ Serious concerns for indirectness due to study population being limited geographically. ^c.^ Serious concerns for indirectness due to study population being limited to one gender. ^d.^ Very serious concerns due to definitely high risk of bias in at least one domain in each study. ^e.^ Serious concerns for imprecision as the 95% CI is wide and crosses 1. ^f.^ Serious concerns for probably high risk of bias in multiple domains (selection bias, incomplete outcome data, selective reporting of outcome, differences in numerator and denominator and outcome misclassification). ^g.^ Serious concern for indirectness due to occupational groups being very specific and thus limited. * The risk in the intervention group (and its 95% confidence interval) is based on the assumed risk in the comparison group and the relative difference (and its 95% CI). CI: Confidence interval; OR: Odds ratio. GRADE quality of evidence ratings. High quality: further research is very unlikely to change our confidence in the estimate of effect. Moderate quality: further research is likely to have an important impact on our confidence in the estimate of effect and may change the estimate. Low quality: further research is very likely to have an important impact on our confidence in the estimate of effect and is likely to change the estimate. Very low quality: we are very uncertain about the estimate.

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
