# Peer review of "Health Services Use and Health Outcomes among Informal Economy Workers Compared with Formal Economy Workers: A Systematic Review and Meta-Analysis"

_ijerph, 2021, doi:10.3390/ijerph18063189_

Round 1
Reviewer 1 Report
The article is written in a very straightforward manner; however I consider that it needs some minor adjustments before publishing
- I consider that the quality of figures 2, 3, 4, 5 should be increase.
- In the conclusions, I think that it would be good to broaden the discussion on the reasons why people in the informal sector tend to have greater mental health problems.
- In the same vein, it would be beneficial, if possible, to include some information about the impact of COVID-19 on the health services for informal and formal workers, as SARS-COV2 pandemic has been devastating for the informal sector. If not possible, I suggest including a paragraph in the limitations.
- I also consider that it is necessary to give a clear definition in the introduction of “informal workers” as it is necessary to separate this kind of workers from the workers in critical conditions of occupation.
Author Response
|
|
Comment |
Response to reviewer |
|
1 |
I consider that the quality of figures 2, 3, 4, 5 should be increase. |
This has been attended to. |
|
2 |
In the conclusions, I think that it would be good to broaden the discussion on the reasons why people in the informal sector tend to have greater mental health problems |
In the discussion on page 13 line 358-360 the possible reasons for mental health problems in the informal workers is described. |
|
3 |
In the same vein, it would be beneficial, if possible, to include some information about the impact of COVID-19 on the health services for informal and formal workers, as SARS-COV2 pandemic has been devastating for the informal sector. If not possible, I suggest including a paragraph in the limitations |
The authors agree that the risks of exposure, access and use of health services is a major problem for informal economy workers, particularly during the COVID 19 pandemic. (ILO brief on: COVID-19 crisis and the informal economy Immediate responses and policy challenges (https://www.ilo.org/wcmsp5/groups/public/---ed_protect/---protrav/---travail/documents/briefingnote/wcms_743623.pdf).
However, a comparison between the informal workers and formal workers’ health service use has not been conducted to inform this research. Also, there is anecdotal evidence that formal economy workers have experienced reduced health service access as well. Thus it is not included. |
|
4 |
I also consider that it is necessary to give a clear definition in the introduction of “informal workers” as it is necessary to separate this kind of workers from the workers in critical conditions of occupation |
Informal workers has been defined on page 2, line 92-95 |
Reviewer 2 Report
This appears to be the first systematic review based on PRISMA synthesising evidence on health services use and work-related health outcomes by economic formality.
Overall the work is presented well - though the figure quality needs significantly improved and the conclusion is far too brief for a paper that delves into so much detail. A proper reflection on the key findings would therefore be needed to improve the paper.
Author Response
|
|
Comment |
Response to reviewer |
|
1 |
The figure quality needs significantly improved |
This has been attended to. |
|
2 |
The conclusion is far too brief for a paper that delves into so much detail |
Conclusion improved. Page 13 line 388-396. |
|
3 |
A proper reflection on the key findings would therefore be needed to improve the paper |
Additional information has been added to the discussion and conclusion From page 11 to 13 , line 312 to line 396 |
Reviewer 3 Report
The utility of this study is the demonstration that there is a lack of studies focusing on the informal work sector and health outcomes. Thank you for your submission.
- The focus of the introduction is on occupational hazards and risks. However, the paper quickly turns to access to health services. With such a small number of studies, how does this focus inform recommendation to reduce occupational health and safety risks in the informal sector. From Line 48 of the manuscript "Consequently, their working environments may have poor hazard control, and they may experience greater occupational health risks than their counterparts in the formal economy." There is no discussion of hazard controls in this sector leading to health outcomes or access to health services.
Line 89: Why less or equal to 16 years old? Give a reason for this as many countries all formal and informal work at 14 years.
Section 2.2.1: Need for more background and results on workers aged 16-17 years. Can any comparisons between formal and informal sectors be made for workers in this age group?
Line 95: "Expert group" - please explain if you are going to call them experts. Need for background and experience. Otherwise, call them researchers.
Line: 95: How were the outcomes selected? By consensus, previously published documents? There is no reference to the Sustainable Development Goals. I'm not sure where these outcomes originated. Need more background on this method criteria.
Table 1: How were these outcomes deemed work-related? Example - how did the articles ensure that TB was from a work-related exposure or a nonwork-related exposure. Many of these outcomes (beside workplace fatalities and injuries) would be difficult to determine work-relatedness.
Section 3.2: Several selected studies included have non-statistically significant outcomes (study 8, 9, 12, 1) with not significance reported for studies 5-7.
Not sure what table 3 adds to the study. Keep just as a reference to the supplemental tables.
Need better explanation of "Risk of bias". Confusing to the reader.
Lines 200-204: Need better explanation of "downgrading". What is the purpose? Demonstrate with other studies.
Author Response
|
|
Comment |
Response to reviewer |
|
1 |
The focus of the introduction is on occupational hazards and risks. However, the paper quickly turns to access to health services. With such a small number of studies, how does this focus inform recommendation to reduce occupational health and safety risks in the informal sector. From Line 48 of the manuscript "Consequently, their working environments may have poor hazard control, and they may experience greater occupational health risks than their counterparts in the formal economy." There is no discussion of hazard controls in this sector leading to health outcomes or access to health services. |
There is virtually no information on hazard control such as the use of PPE, administrative or environmental controls in the informal economy. Due to the informality of employment and the lack of monitoring to assess if controls are in place it is a possibility that their risks of exposure are high. This review illustrates the lack of quality research comparing the informal and formal sector health outcomes and health service use. It provides evidence that health system improvements are required in order for the informal economy workers to easily access and use health services with minimal impact on their income generation. In addition, a focus by the health systems on mental health is essential and should be included in primary health care services.
|
|
2 |
Line 89: Why less or equal to 16 years old? Give a reason for this as many countries all formal and informal work at 14 years |
The use of 16 years as the lower age limit for a working population was based on the ILO criterion, although the working age can vary between countries. (Page 2 line 94) We did not include any studies that reported on child labour. |
|
3 |
Section 2.2.1: Need for more background and results on workers aged 16-17 years. Can any comparisons between formal and informal sectors be made for workers in this age group? |
There were only two studies that reported in this age group, however the results were not stratified by age and thus a comparison cannot be made. |
|
4 |
Line 95: "Expert group" - please explain if you are going to call them experts. Need for background and experience. Otherwise, call them researchers. |
Added the reason for describing the group as experts- page 3, line 103-104. |
|
5 |
Line: 95: How were the outcomes selected? By consensus, previously published documents? There is no reference to the Sustainable Development Goals. I'm not sure where these outcomes originated. Need more background on this method criteria. |
The expert group, based on their knowledge and experience in the field, selected the outcomes. Some of the outcomes are based on the SDG’s. Those that were not in the SDG’s were found to be most prevalent in an occupational setting – Musculoskeletal disorders and Noise induced hearing loss- compared to the many other occupational diseases. Page 3 lines 106-107 |
|
6 |
Table 1: How were these outcomes deemed work-related? Example - how did the articles ensure that TB was from a work-related exposure or a nonwork-related exposure. Many of these outcomes (beside workplace fatalities and injuries) would be difficult to determine work-relatedness |
This is true, however there are certain occupations that have a higher risk for some diseases e.g. TB in the mining industry. In addition, even though the exposure may not have occurred in the workplace – the outcomes of ill health affecting the informal vs the formal sector still has an impact on the worker, the industry and the health system. Thus the purpose of the review was to assess the differences in outcomes in the two groups and not the exposures. |
|
7 |
Section 3.2: Several selected studies included have non-statistically significant outcomes (study 8, 9, 12, 1) with not significance reported for studies 5-7. |
Studies for inclusion in systematic reviews are selected by pre-determined suitability criteria which purposefully do not include the statistical significance of the findings, either positive or negative.
Studies 5-7 do not have 95% CI or p values This has been added as a footnote to the table 2. |
|
8 |
Not sure what table 3 adds to the study. Keep just as a reference to the supplemental tables. |
Table 3 is a heat map for the risk of bias in a systematic review. The authors would prefer to retain this in the main body of the paper. This reporting follows the layout for the majority of systematic reviews published internationally. |
|
9 |
Need better explanation of "Risk of bias". Confusing to the reader. |
Explained on page 4, line 131-141 in the Materials and Methods. |
|
10 |
Lines 200-204: Need better explanation of "downgrading". What is the purpose? Demonstrate with other studies. |
Downgrading or upgrading of evidence is explained on page 4, line 156 to 166. This is a standard way of assessing the quality of evidence in a systematic review with meta analyses. If there are concerns about a study quality it is downgraded. This cannot be compared to other studies in this context. |
In addition to responding to the reviewers’ comments, we have:
- Further improved the framing of differences in health services use and health outcomes in light of the social determinants of health and health equity, including added references to key documents supporting such framing.
- Further improved the reporting of our findings, including ensuing our conclusions (where we had at least low quality of evidence) are made fully explicit at first reporting.
- Further aligned and improved the risk of bias and summary of evidence tables, ensuring they follow the latest international reporting guidelines for these standard tables for systematic reviews.
- Conducted another technical edit of the manuscript for both concision and style.
Round 2
Reviewer 3 Report
All of my comments were addressed satisfactorily.